# Some Critical Remarks about Mathematical Model Used for the Description of Transport Kinetics in Polymer Inclusion Membrane Systems

**DOI:** 10.3390/membranes10120411

**Published:** 2020-12-10

**Authors:** Piotr Szczepański

**Affiliations:** Faculty of Chemistry, Nicolaus Copernicus University in Toruń, 7 Gagarina Street, 87–100 Toruń, Poland; piotrsz@umk.pl

**Keywords:** membrane transport models, kinetic models, model fitting, polymer inclusion membrane transport

## Abstract

Two kinetic models which are applied for the description of metal ion transport in polymer inclusion membrane (PIM) systems are presented and compared. The models were fitted to the real experimental data of Zn(II), Cd(II), Cu(II), and Pb(II) simultaneous transport through PIM with di-(2-ethylhexyl)phosphoric acid (D2EHPA) as a carrier, o-nitrophenyl octyl ether (NPOE) as a plasticizer, and cellulose triacetate (CTA) as a polymer matrix. The selected membrane was composed of 43 wt. % D2EHPA, 19 wt. % NPOE, and 38 wt. % CTA. The results indicated that the calculated initial fluxes (from 2 × 10^−11^ up to 9 × 10^−10^ mol/cm^2^s) are similar to the values observed by other authors in systems operating under similar conditions. It was found that one of the most frequently applied models based on an equation similar to the first-order chemical reaction equation leads to abnormal distribution of residuals. It was also found that application of this model causes some problems with curve fitting and leads to the underestimation of permeability coefficients and initial maximum fluxes. Therefore, a new model has been proposed to describe the transport kinetics in PIM systems. This new model, based on an equation similar to the first-order chemical reaction equation with equilibrium, was successfully applied. The fit of this model to the experimental data is much better and makes it possible to determine more precisely the initial maximum flux as the parameter describing the transport efficiency.

## 1. Introduction

A simple kinetic model equation proposed by Danesi [1,2] is one of the most frequently applied in description of various substance transports through supported liquid membranes (SLMs) and polymer inclusion membranes (PIMs). This equation was used for the transport description of metal cations [3,4,5,6,7], inorganic anions [8], as well as a variety of organic substances such as phenol [9], carboxylic acids [10], and herbicide [11]. This useful model can be easily derived from Fick’s first law after assuming linear concentration gradients, steady-state diffusion with fast interfacial reaction, low concentration of substance in the feed (donor) solution, and much larger distribution constant at the feed/membrane interface than that at the opposite side [1]. Accordingly, the molar flux through the membrane at any time is given by:(1)$$J=-\frac{V_{f}}{A_{}}\;\frac{dc_{f}}{dt}=\frac{K\bar{D}}{l}c_{f}.$$

In the above equation, *V_f_* denotes the volume of the feed solution (m^3^), *A*—an area of membrane (cm^2^), *c_f_*—the concentration of a substance at the given time *t* in the feed solution [mol/cm^3^], *K*—distribution constant, *l*—membrane thickness [cm], and $\bar{D}$—the diffusion coefficient of transported substance in the membrane [cm^2^/s].

This model is primarily applied for quick calculation of the initial flux (*J_i_*, mol/cm^2^s)—a parameter which describes the transport efficiency and corresponds to the maximum flux observed in the membrane system:(2)$$J_{i}=P\;c_{f,t=0},$$
in which *c_f_*_, *t* = 0_ denotes the initial concentration of a substance in the feed solution [mol/cm^3^], and *P* is the permeability coefficient [cm/s]. The permeability coefficient (*P*) can be easily calculated as the slope of a linear dependence, which is the integrated form of Equation (1):(3)$$\ln\frac{c_{f}}{c_{f,t=0}}=-P\frac{A}{V_{f}}t.$$

Equation (3), which is similar to the equation describing first-order reaction kinetics, is also applied for transport evaluation in supported liquid membrane (SLM) systems [12]. Equation (3) can be easily converted to the following relationship:(4)$$\ln\left(1-\frac{V_{s}\;c_{s}}{V_{f}\;c_{f,\;t=0}}\right)=-P\frac{A}{V_{f}}t,$$
which describes the change of transported substance concentration in the stripping solution (*c_s_*), vs. denotes the volume of the stripping solution [cm^3^].

It should be noted that Equation (3) is frequently applied for the description of membrane transport in systems in which some of the above mentioned assumption are not fulfilled. First of all, this applies to systems in which the transported substance may diffuse back into the feed solution i.e., the transported substance does not undergo an irreversible reaction in the receiving (stripping) solution. In such systems, it cannot be assumed that the distribution constant at the feed/membrane interface is much larger than that at the membrane/stripping interface. Accordingly, the lack of model fit is particularly apparent in the ln(*c_f_*/*c_f_*_,*t* = 0_) vs. time plots on which the experimental points deviate from the linear model [3,10,13,14,15,16].

Because some of the experimental results indicated that the dependence of ln(c*_f_*/c*_f_*_0_) vs. time may not be linear, in this manuscript, a modified equation for the transport description will be proposed. Similar equations were applied for the description of the uptake and release of alkali metal cations in a bulk liquid membrane system [17]. The proposed model can also be derived from the kinetic model of the first-order reversible reaction:(5)$$c_{f}\overset{k_{1}}{\underset{k_{2}}{⇄}}c_{s},$$
where *c_f_* and *c_s_* denote substance concentration in the feed and stripping solution, respectively. The rate of change in concentrations can be expressed by the following differential equations:(6)$$\frac{dc_{f}}{dt}=-k_{1}c_{f}+k_{2}c_{s},$$
(7)$$\frac{dc_{s}}{dt}=k_{1}c_{f}-k_{2}c_{s}.$$

Taking into account the volume of the feed (*V_f_*) and stripping (*V_s_*) solution, the membrane surface area (*A*), and assuming that the concentration in the feed and the stripping solution is allowed to approach equilibrium, the respective flux equation derived from Equation (5) can be integrated to:(8)$$\ln\frac{c_{f}-c_{f\inf}}{c_{f,t=0}-c_{f\inf}}=-\frac{P_{1}Ac_{f,t=0}}{V_{f}(c_{f,t=0}-c_{f\inf})}t,$$
for the feed, and:(9)$$\ln\frac{c_{s\inf}-c_{s}}{c_{s\inf}}=-\frac{P_{1}Ac_{f,t=0}}{V_{s}c_{s\inf}}t,$$
for the stripping solution.

In the above equations, *P*_1_ denotes the permeability coefficient [cm/s], *c_f_*
_inf_ and *c_s_*
_inf_—the concentration of a substance in the feed (*f*) and stripping (*s*) solution at infinite time i.e., the equilibrium concentration [mol/cm^3^], while *c_f_* and *c_s_*—the concentration of the feed and stripping solution [mol/cm^3^], respectively. The correctness of the *P*_1_ value can easily be verified by calculating the slope of the linear dependencies from both Equations (7) and (8). The difference between these two values may result from the retention of a substance in the membrane or measurement errors in the determination of the transported substance concentration.

For initial condition *t* = 0 and *c_f_* = *c_f_*_0_, the initial maximum flux (*J_i,eql_*, [mol/cm^2^s]) can be calculated from the following well-known relationship:(10)$$J_{i,eql}=-\frac{V_{f}}{A}\frac{dc_{f}}{dt}=P_{1}c_{f0}.$$

Equation (8) can be easy simplified to the relationship presented by Equation (3) by assuming that the concentration of a substance in the feed solution at infinite time goes to zero (*c_f_*_inf_ → 0). Another simplification of Equation (8) can be obtained assuming that *V_f_* = vs. and *c_f_*_inf_ = *c_f_*_,*t* = 0_/2. Such conditions indicate the membrane system in which a simple diffusion or carrier-mediated simple diffusion process occurs. The respective equation takes the following form:(11)$$\ln\frac{2c_{f}-c_{f,t=0}}{c_{f,t=0}}=-\frac{2P_{d}A}{V_{f}}t,$$
in which *P*_d_ denote the diffusive permeability coefficient [cm/s]. For instance, Equation (11) was applied for characterization of NaCl diffusive permeability through PIM containing Aliquat 336 [18] or organic acids transport across PIM in which 1-alkylimidazols and TOA were applied as a carrier [19,20].

It should be noted, that the application of different models may lead to obtaining different values of *P* and *J_i_*, and therefore unification of the method of their calculation is extremely important. The simplicity of the equation of a given model cannot be the only criterion for its application. Consequently, the present paper is focused on simple kinetic models used for membrane transport evaluation in PIM systems and characterizes some problems arising during model parameter calculation. A new model is proposed to evaluate the transport kinetics in PIM systems. A detailed description of the new model as well as advantages and drawbacks over the typical one is presented.

## 2. Experimental

In order to evaluate the correctness of the models described above, various transport experiments were performed in a membrane system in which polymer inclusion membranes were applied. The transport experiments were performed in a permeation setup presented schematically in Figure 1.

The feed solution was composed of Zn(II), Cd(II), Cu(II), and Pb(II) nitrates (Sigma-Aldrich, Stainheim, Germany, reagent grade, purity ≥98%) dissolved in double distilled water (200 cm^3^) with an initial concentration range from 5 × 10^−4^ to 0.01 M. As the stripping phase, a nitric acid solution of 0.5 M concentration and volume of 200 cm^3^ was used. Cellulose triacetate (CTA, Acros Organics, Geel, Belgium) as a polymer matrix, o-nitrophenyl octyl ether (NPOE, Alfa Aesar, Heysham, Lancashire, United Kingdom, 98%) as a plasticizer, and di-(2-ethylhexyl) phosphoric acid (D2EHPA, Aldrich, Stainheim, Germany, 97%) as a carrier were applied as the membrane components. The surface membrane area was equal to 17 cm^2^.

### 2.1. Membrane Preparation

PIMs were prepared as flat membranes by a solution casting and solvent evaporation technique at room temperature [21]. A solution of CTA, carrier and the plasticizer in dichloromethane (Riedel-de Haën, purity ≥99.9%) were prepared and poured into a Petri dish (7 cm diameter). After evaporation of the organic solvent (24 h), the membrane was immersed for 12 h in distilled water to swell and used in a transport of metal ions. The composition of the membrane was the same for all of the experiments i.e.,: 43 wt. % D2EHPA, 19 wt. % NPOE, and 38 wt. % CTA. The concentrations of D2EHPA and NPOE selected for the experiments correspond to the maximum fluxes of ions observed by other authors in the similar systems [22,23]. The total membrane weight after solvent evaporation equaled (0.312 ± 0.008) g and its thickness was (0.0568 ± 0.0042) mm.

### 2.2. Concentration Analysis

Concentration of the aqueous solutions was carried out by the standard atomic absorption method using the SPECTRAA 20ABQ Varian spectrophotometer.

## 3. Results

Typical results of the feed and stripping solution concentration vs. time dependences for the system with D2EHPA as a carrier are presented in Figure 2.

The results indicated that transport of all ions occurs under non-stationary conditions. Therefore, Equation (3) seems to be the appropriate choice for permeability coefficient calculation. The fit of the model to the experimental data in the form of the ln(*c*/*c_0_*) = *f*(*t*) dependence for Zn(II), Cd(II), Pb(II), and Cu(II) ions is shown in Figure 3a.

The results presented in Figure 3a indicated that the model fitted the data relatively well with determination coefficients equaling 0.9811, 0.9890, 0.9912, and 0.9663 for Zn(II), Cd(II), Pb(II), and Cu(II) ions, respectively. The model represented by Equation (3) is very simple and easy to apply, although it should be remembered that in the case of linear regression, residuals should be independent of each other and normally distributed with a mean of zero [24]. Residuals are the vertical distances between data points and the regression line and play an essential role in regression diagnostics. Without a thorough examination of residuals, no analysis is complete.

As an example of a residuals analysis and initial flux calculation method, the results of Zn(II) ions transport will be described in more detail. Scatter plots of residuals for Zn(II) ions dependence (Figure 3b) show that they are not normally distributed. Therefore, it should be concluded that this equation (Equation (3)) is inapplicable i.e., the assumption of linearity is incorrect and the model is inadequate. It should be noted that incorrect distribution of residuals was also observed in the results presented by other authors for the transport of various substances in PIM and SLM systems [3,10,13,14,15,16]. In order to avoid this problem, a number of researchers select initial points for which the linearity of Equation (3) and normality of distribution is maintained. For example, it was found that for the results presented in Figure 3, the linearity is observed for *n* = 6 initial experimental points. The results of model fit and residual plot are compared in Figure 4.

The residual plot (Figure 4b) shows a fairly random pattern. Moreover, it can be confirmed by normality tests (e.g., Kolmogorov–Smirnov test, *p* = 0.2) that residuals are normally distributed, indicating that the evaluated linear model for *n* = 6 experimental points is adequate. The randomness of distribution can be confirmed by an appropriate statistical test (Runs test). However, this test is impossible to perform owing to the insufficient number of experimental points (required *n* = 8).

The different slope values calculated from the data shown in Figure 3 and Figure 4 lead to different values of the permeability coefficient and, consequently, to differences in the fit of the model. The model fit for the values of *P* = 1.464(0.042) × 10^−4^ cm/s (for *n* = 10) and *P* = 1.78(0.03) × 10^−4^ cm/s (for *n* = 6) is shown in Figure 5, respectively.

The differences in fit are significant and clearly show that the best fit is observed for *P* = 1.78 × 10^−4^ cm/s. The calculated initial flux value is therefore 21% higher than those calculated in the case of the model in which all experimental points were included. It means that these values are dependent on the number of points which are taken into account in model calculations. The effects of the number of experimental points in regression analysis on permeability coefficient (*P*) and initial flux (*J_i_*) values calculated from Equations (2) and (3) for *n* = 2 to 10, is presented in Table 1. Moreover, in Table 1, the sum of the squared estimate of errors (SSE) for nonlinear feed solution concentration vs. time dependence (*c_f_* = *f*(*t*)) was calculated and compared.

Because of specific residuals distribution (see Figure 3b), the values of permeability coefficient and initial flux increases with the decrease in the number of experimental points up to *n* = 3. However, the lowest value of SSE clearly indicates that the best fit (see Figure 5b) is observed for *P* = 1.78 × 10^−4^ cm/s. Despite the fact that determination coefficients increase with the decrease of *n* (as well as degree of freedom), it should be remembered that the statistical power of regression models decreases. The presented results indicate that the selection of an appropriate number of initial experimental points is not simple. It requires the application of appropriate statistical tests and calculations in order to determine the best model describing the relationship between *c_f_* and time. It should be noted that most authors do not present the fit of the model to the experimental data for *c_f_* vs. time dependence, but only the fit of ln(*c*/*c*_0_) vs. time relationship. Only the presentation of both fits for *c_f_* = *f*(*t*) and *c_s_* = *f*(*t*) dependences can unequivocally confirm the validity of the model and prove the lack of accumulation of the transported substance in the membrane (see Figure 5b). Nevertheless, the fit of the model, especially for *c_f_* = *f*(*t*) plot and time higher than 40 h, is still unsatisfactory and suggests that the concentration is reaching equilibrium.

In some cases, the method for selection of initial experimental points fails. For instance, in Figure 6, the experimental results of Cu(II) ion transport through PIM are demonstrated. In this system, the initial feed solution concentration of Zn(II), Cd(II), Pb(II), and Cu(II) ions was 5 × 10^−4^ M. In all the studied systems with D2EHPA as a carrier, Cu(II) ions are not preferentially transported and selectivity decreases in order: Zn ≥ Pb > Cd > Cu.

Two different model fitting dependences are presented in Figure 6. The first one (solid line), calculated using permeability coefficient value evaluated from all of the experimental points (*P* = 7.282 × 10^−5^ cm/s) and the second one, with *P* value calculated from initial experimental points (*n* = 4) for which ln(*c*/*c*_0_) vs. time relationship is linear (*P* = 1.144 × 10^−4^ cm/s, dashed line). The results clearly indicate that the model fits are not satisfactory. For the first dependence (solid line), the flux value is underestimated, while for dependence represented by dashed line, the initial flux value is estimated much better. In spite of that, for time of transport exceeding 20 h, this model fit is evaluated with an unacceptably large error (dashed line). Finally, it should be mentioned that some researchers applied Equation (3) for the kinetics transport description in PIM systems in which accumulation of a transported substance in the membrane phase is significant. In such a case, this equation describes only the sorption process into the membrane, not the transport through the membrane. Therefore, another model should be applied, e.g., the model derived from the kinetic laws of two consecutive irreversible (or reversible) first-order reactions [25,26]. Such type of models are however beyond the scope of this manuscript. 

The problems mentioned above with model fitting using the equation proposed by Danesi [1,2] can be solved by the application of the model presented by Equations (8) and (9). Examples of the model fit to the experimental data are shown in Figure 7.

The results of the fitted model (line) calculated from Equations (8) and (9) indicated a very good fit to the experimental data. Moreover, the performed statistical tests (runs test and Shapiro–Wilk test) confirmed that the runs are random and residuals are normally distributed.

Figure 8 shows the correlation between the permeability coefficient values calculated from Equation (3) as well as Equations (8) and (9) in which all of the experimental data were taken into account. The results indicated that most of the permeation coefficient values calculated from Equation (3) are underestimated. This means that the application of the new model proposed in this manuscript leads to higher values of *P* and *J_i_* compared to the model proposed by Danesi [1,2]. The relative error between these calculated values ranges from 0 up to 47%.

The value of the maximum flux depends on many operational conditions e.g., the concentration and the type of the feed and the stripping solution. In a particular way, this value depends on the membrane composition, its thickness, and the type of carrier. In order to determine the influence of the membrane composition on the concentration vs. time dependence, several additional experiments were carried out. The results confirm that the model proposed by Danesi [1,2] can be applied only for description of the transport kinetics for substances with the highest selectivity (i.e., the highest flux), whereas the model represented by Equations (8) and (9) is more flexible and can be applied for all of the transported substances (ions) regardless of the selectivity. This topic, however, is beyond the scope of this manuscript and will be considered and discussed in the subsequent manuscript.

One of the main disadvantages of the proposed model is the necessity to adjust two parameters, i.e., *P*_1_ and *c_f inf_* (or *c_s inf_*) values by iterative calculations, especially in the case when *c_f inf_* or *c_s inf_* values are experimentally inaccessible i.e., if the concentration in the respective phases has not reached equilibrium. It is worth noticing that these calculations can be carried using the Solver add-in in MS Excel, or by fitting the numerical solution of differential equations (Equations (6) and (7)) to the experimental data, e.g., in the Berkeley Madonna program (ordinary differential equations (ODEs) solver). In both methods, it is possible (and advisable) to use in calculations, the dependence of concentration changes in both the feed and the stripping solution. As a result, the number of experimental points (degrees of freedom) used to estimate model parameters increases significantly and ultimately increases the accuracy of the model prediction.

## 4. Conclusions

The presented results indicated that the application of a model based on an equation similar to the first-order chemical reaction equation (Equation (3)) is severely limited. As previously stated, in some cases, because of abnormally distributed residuals, the assumption of linearity is incorrect and the model is inadequate. Only for systems in which the transported substance reacts immediately in the stripping solution, creating a product which is not transported through the membrane, does the application of Equation (3) seem to be justified. In the case of the systems under investigation, satisfactory results are observed only for description of the transport kinetics for substances with the highest selectivity (in the discussed example it concerns Zn(II) and Pb(II) ions) and for permeability coefficients calculated from selected initial points.

In contrast, the model proposed in this paper, based on an equation similar to the first-order chemical reaction equation with equilibrium, is more flexible and provides the best nonlinear fit to the experimental data. The application of this model leads to more accurately estimated (and larger) values of the permeation coefficients and initial maximum fluxes. The fact that in the case of the proposed model two parameters should be estimated simultaneously is an unquestionable difficulty. However, an appropriate computer software provides a very simple way to solve this problem.

## Figures and Tables

**Figure 1 membranes-10-00411-f001:**
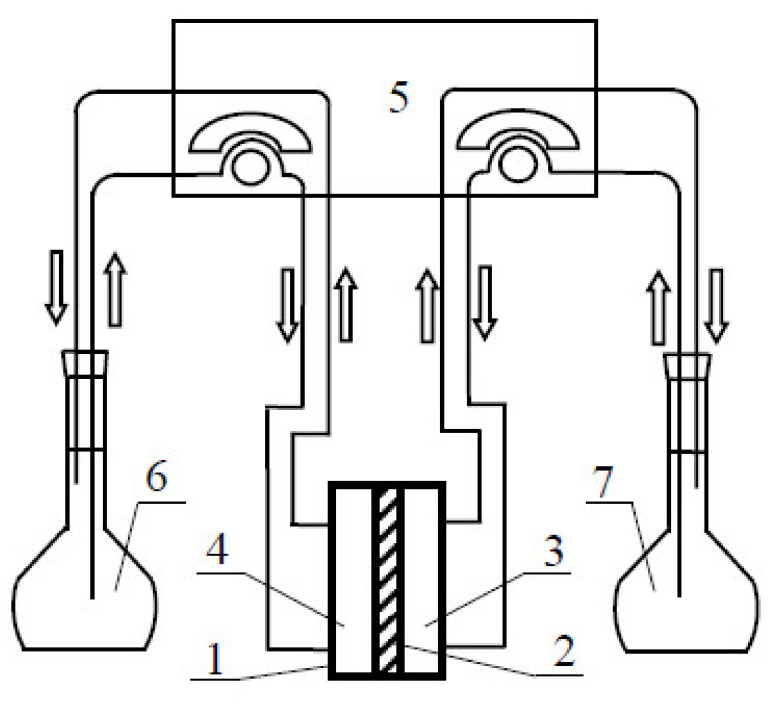
Scheme of the permeation system used in the experiments: permeation cell (1), membrane (2), stripping phase compartment (3), feed phase compartment (4), peristaltic pump (5), feed solution reservoir (6), stripping solution reservoir (7).

**Figure 2 membranes-10-00411-f002:**
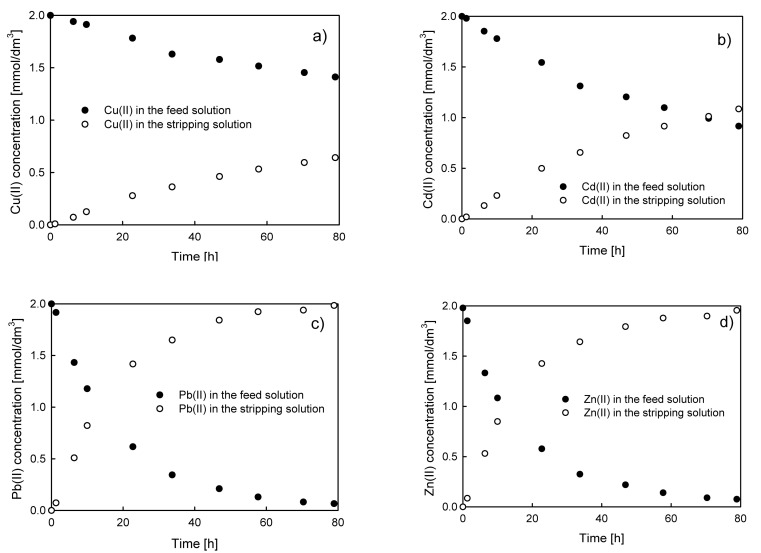
Typical experimental results of (**a**) Cu(II), (**b**) Cd(II), (**c**) Pb(II), and (**d**) Zn(II) ions transport through polymer inclusion membrane (PIM) composed of cellulose triacetate (CTA) as a polymer matrix, di-(2-ethylhexyl) phosphoric acid (D2EHPA) as a carrier, and o-nitrophenyl octyl ether (NPOE) as a plasticizer. Feed solution concentration 0.002 M, stripping solution (HNO_3_) concentration 0.5 M.

**Figure 3 membranes-10-00411-f003:**
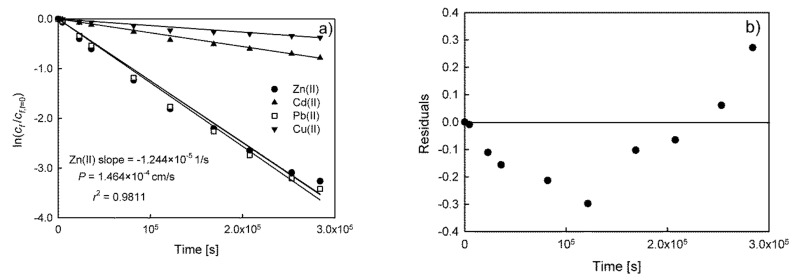
The results of model fit (Equation (3)) to the experimental data (**a**) for Zn(II), Cd(II), Pb(II), and Cu(II) ions, as well as residual plot (**b**) for Zn(II) ions transport through PIM with D2EHPA as a carrier.

**Figure 4 membranes-10-00411-f004:**
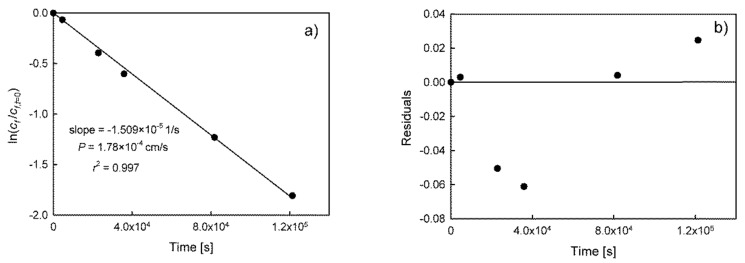
The results of model fit (Equation (3)) to the experimental data for 6 initial points (**a**) and residual plot (**b**) for Zn(II) transport through PIM with D2EHPA as a carrier.

**Figure 5 membranes-10-00411-f005:**
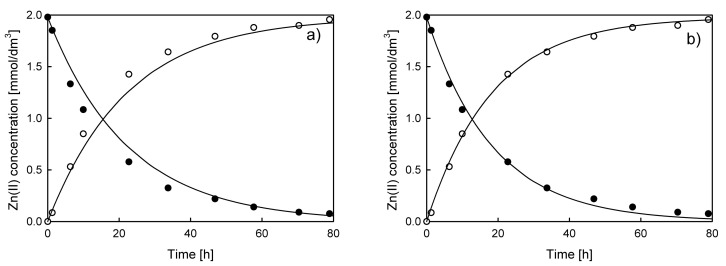
The model fit to the experimental data of Zn(II) ions transport through PIM with D2EHPA as a carrier, using *P* = 1.464 × 10^−4^ cm/s (**a**) and *P* = 1.78 × 10^−4^ cm/s (**b**).

**Figure 6 membranes-10-00411-f006:**
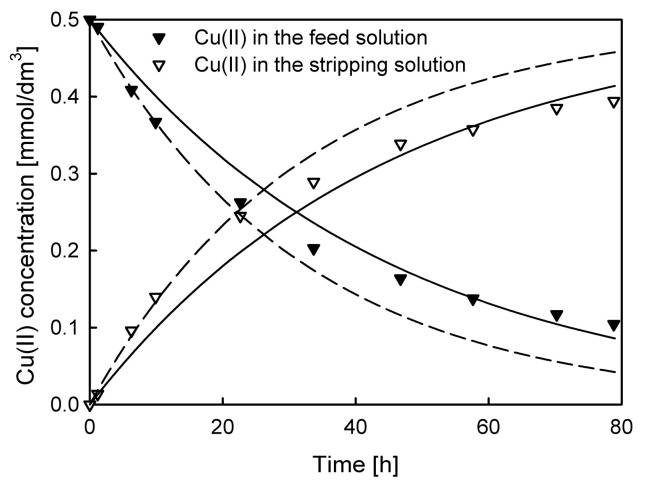
Experimental results (points) and model fitting for *n* = 10 (line) and *n* = 4 (dashed line) for Cu(II) ions transport through PIM with D2EHPA as a carrier, 0.5 M HNO_3_ as the stripping solution, and 5 × 10^−4^ M Zn(II), Cd(II), Pb(II), and Cu(II) nitrates as the feed solution.

**Figure 7 membranes-10-00411-f007:**
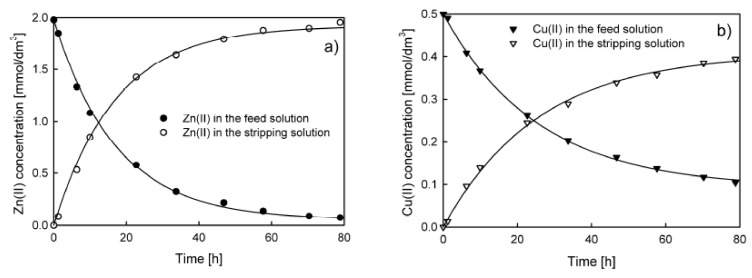
Examples of the model fit (Equations (8) and (9)) to the experimental data of Zn(II) and Cu(II) ions transport through PIM with D2EHPA as a carrier. Concentration of the stripping solution—0.5 M HNO_3_, the feed solution concentration 2 × 10^−3^ M (**a**) and 5 × 10^−4^ M (**b**) of Zn(II), Cd(II), Pb(II), and Cu(II) nitrates.

**Figure 8 membranes-10-00411-f008:**
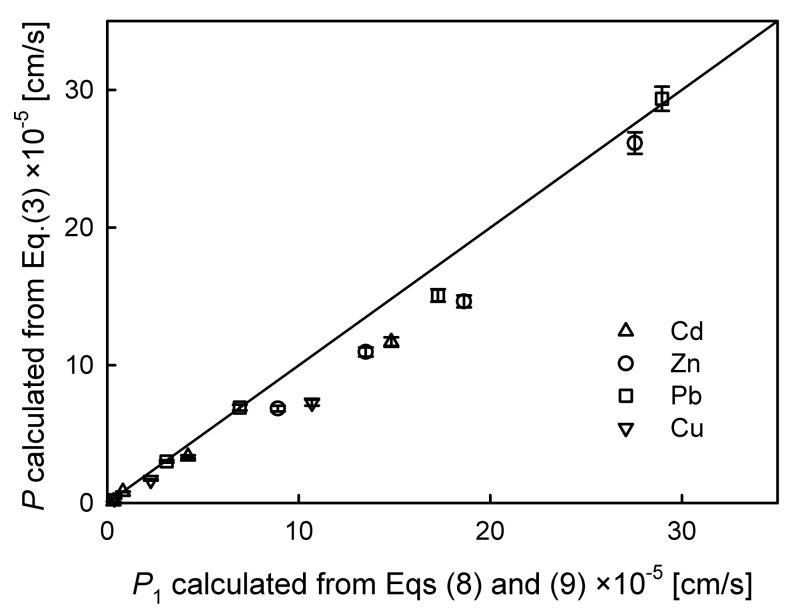
The correlation plot between permeability coefficients calculated from Equation (3) as well as Equations (8) and (9). Permeability coefficients of Zn(II), Cd(II), Pb(II), and Cu(II) in systems with D2EHPA as a carrier, 0.5 M HNO_3_ as the stripping solution, and an initial feed solution concentration from 5 × 10^−^^4^ to 0.01 M.

**Table 1 membranes-10-00411-t001:** The influence of number of experimental points in linear regression analysis on permeability coefficient (*P*), initial flux (*J_i_*) value evaluated from Equations (2) and (3), and determination coefficient (*r*^2^) as well as the sum of squared estimate of errors (SSE) for *c_f_* = *f*(*t*) dependence.

Number of Experimental Points in Regression Analysis	Permeability Coefficient, *P* [cm/s] *	Initial Flux, *J_i_* [mol/cm^2^s] *	Determination Coefficient, *r*^2^	SSE ** *n* = 10 × 10^8^
2	1.7 × 10^−4^	3.4 × 10^−10^	1	2.55
3	2.02(0.05) × 10^−4^	4.0(0.1) × 10^−10^	0.9989	4.28
4	1.990(0.024) × 10^−4^	3.98(0.05) × 10^−10^	0.9996	3.74
5	1.82(0.05) × 10^−4^	3.6(0.1) × 10^−10^	0.9974	2.12
6	1.78(0.03) × 10^−4^	3.55(0.06) × 10^−10^	0.9987	**2.11**
7	1.644(0.052) × 10^−4^	3.29(0.11) × 10^−10^	0.994	3.34
8	1.579(0.045) × 10^−4^	3.16(0.09) × 10^−10^	0.9944	4.79
9	1.521(0.041) × 10^−4^	3.041(0.082) × 10^−10^	0.9943	6.65
10	1.464(0.042) × 10^−4^	2.927(0.083) × 10^−10^	0.9929	9.11

* standard deviation in parentheses. ** the lowest SSE bolded.

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
