# Peer review of "Some Critical Remarks about Mathematical Model Used for the Description of Transport Kinetics in Polymer Inclusion Membrane Systems"

_membranes, 2020, doi:10.3390/membranes10120411_

Round 1
Reviewer 1 Report
Comments and Suggestions on Membrane-994564: Some critical remarks about mathematical model used for description of transport kinetics in polymer inclusion membrane systems
This manuscript is logically organized and provides a novel mathematical model for description of transport kinetics in polymer inclusion membrane systems. Based on equation similar to first–order chemical reaction equation with equilibrium, the new model was applied in a good nonlinear fitting of the experimental data and determined the permeation coefficients and the initial maximum fluxes more accurately compared to the frequently applied model. This work is suggested to be considered to publish in the journal with careful revisions. Following are some comments:
- The whole paper should be presented in a uniform font. (Some sentences are in italics)
2 What’s the type of polymer inclusion membrane, flat, hollow fiber, or else? And the critical specification of this membrane needs to be provided.
3 How is the surface membrane area determined?
4 Line 153-155, What’s the minimum number of the experimental points required in statistical test for confirmation of the randomness of distribution and the reason?
5 Line 191-194, The selectivity (Zn≥Pb>Cd>Cu) can’t be observed from the Fig.6.
Author Response
RESPONSE TO COMMENTS
Reviewer #1
Comments and Suggestions on Membrane-994564: Some critical remarks about mathematical model used for description of transport kinetics in polymer inclusion membrane systems
This manuscript is logically organized and provides a novel mathematical model for description of transport kinetics in polymer inclusion membrane systems. Based on equation similar to first–order chemical reaction equation with equilibrium, the new model was applied in a good nonlinear fitting of the experimental data and determined the permeation coefficients and the initial maximum fluxes more accurately compared to the frequently applied model. This work is suggested to be considered to publish in the journal with careful revisions. Following are some comments:
- The whole paper should be presented in a uniform font. (Some sentences are in italics)
Response: The manuscript was sent as a docx file without any special formatting. Probably during automatic formatting, some sentences were not properly formatted. However in the reviewed version of the manuscript all of these sentences has been corrected (marked in green).
- What’s the type of polymer inclusion membrane, flat, hollow fiber, or else? And the critical specification of this membrane needs to be provided.
R: In experiments a flat type PIMs were used. In order to fulfil the Reviewer #1 expectations the Section 2.1. Membrane separation was changed into (changes are marked in blue):
“PIMs were prepared as flat membranes by a solution casting and solvent evaporation technique at room temperature [18]. A solution of CTA, carrier and the plasticizer in dichloromethane (Riedel–de Haën, purity ³99.9%) were prepared and poured into a Petri dish (7 cm diameter). After evaporation of the organic solvent (24 h) the membrane was immersed for 12 h in distilled water to swell and used in a transport of metal ions. The composition of the membrane was the same for all of the experiments i.e.: 43 wt. % D2EHPA, 19 wt. % NPOE and 38 wt. % CTA. Such composition is typical in studies carried out by other authors [19, 20]. The total membrane weight after solvent evaporation equaled (0,312±0.008) g and its thickness was (0.0568±0.0042) mm.”
- How is the surface membrane area determined?
R: The surface the membrane area is equal to the internal dimensions of membrane system area measured by a calliper. It means that the total membrane area was higher than the active membrane area i.e. 17 cm2.
- Line 153-155, What’s the minimum number of the experimental points required in statistical test for confirmation of the randomness of distribution and the reason?
R: According to the runs test, n=8 experimental points are required; therefore an additional explanation was added on page 6 lines 159-160:
“However this test is impossible to perform owing to the insufficient number of experimental points (required n=8).”
- Line 191-194, The selectivity (Zn≥Pb>Cd>Cu) can’t be observed from the Fig.6.
R: This sentence was changed into more accurate:
“In all the studied systems with D2EHPA as a carrier, Cu(II) ions are not preferentially transported and selectivity decreases in order: Zn≥Pb>Cd>Cu.”
Reviewer 2 Report
The manuscript titled “Some critical remarks about mathematical model used for description of transport kinetics in polymer inclusion membrane systems” by Piotr SzczepaÅ„ski, is an interesting compilation of various works related to the polymer inclusion membrane, which considered properly the kinetic and concept of this type of membrane, but for further proceeding some modifications and corrections are required.
- The paper suffers from the poor Language and strongly recommended to be checked and revised by a native speaker.
- The abstract should be revised and summarized by reporting more numerical values.
- The paper should be rich by more relevant and recent studies on PIMs including: history, benefits, ion transport mechanisms, and etc., the following studies can be considered as good candidates: (1. Wang, B., Li, Z., Lang, Q., Tan, M., Ratanatamskul, C., Lee, M., ... & Zhang, Y. (2020). A comprehensive investigation on the components in ionic liquid-based polymer inclusion membrane for Cr (VI) transport during electrodialysis. Journal of Membrane Science, 118016. 2. Shirzad, M., & Karimi, M. (2020). Statistical analysis and optimal design of polymer inclusion membrane for water treatment by Co (II) removal. DESALINATION AND WATER TREATMENT, 182, 194-207., 3. Jha, R., Rao, M. D., Meshram, A., Verma, H. R., & Singh, K. K. (2020). Potential of polymer inclusion membrane process for selective recovery of metal values from waste printed circuit boards: A review. Journal of Cleaner Production, 121621.)
- More explanations about the effect of PIM composition on the final results are required.
- The PIM components (polymer, carrier, and the plasticizer) how have been selected for these experiments? If it has been selected based on a specific reference, please mention it.
- What is “partition coefficient” mentioned in page 2? More explanation is required.
- In section “2.Experimental”, the range of initial ion concentration in feed phase is in the range of 0.0005 to 0.01 M; but the initial ion concentrations showed in Fig. 2 is 2 M!!!! Please explain this discrepancy.
- In section 2.1, total weight of the casted membrane, as well as each PIM components weight must be mentioned.

Author Response
RESPONSE TO COMMENTS
Reviewer #2
The manuscript titled “Some critical remarks about mathematical model used for description of transport kinetics in polymer inclusion membrane systems” by Piotr SzczepaÅ„ski, is an interesting compilation of various works related to the polymer inclusion membrane, which considered properly the kinetic and concept of this type of membrane, but for further proceeding some modifications and corrections are required.
- The paper suffers from the poor Language and strongly recommended to be checked and revised by a native speaker.
Response: The manuscript was carefully checked by a native speaker. All of language changes are marked in red (see List of changes).
- The abstract should be revised and summarized by reporting more numerical values.
R: In my opinion the Reviewer expectation cannot be fulfilled because this manuscript “is focused on simple kinetic models used for membrane transport evaluation in PIM systems and characterizes some problems arising during model parameter calculation.” This work does not concern the system operation optimization and/or the influence of various variables on the transport efficiency and selectivity. Therefore it is hard to decide what kind of numerical values (the best results?) should be presented.
- The paper should be rich by more relevant and recent studies on PIMs including: history, benefits, ion transport mechanisms, and etc., the following studies can be considered as good candidates: (1. Wang, B., Li, Z., Lang, Q., Tan, M., Ratanatamskul, C., Lee, M., ... & Zhang, Y. (2020). A comprehensive investigation on the components in ionic liquid-based polymer inclusion membrane for Cr (VI) transport during electrodialysis. Journal of Membrane Science, 118016. 2. Shirzad, M., & Karimi, M. (2020). Statistical analysis and optimal design of polymer inclusion membrane for water treatment by Co (II) removal. DESALINATION AND WATER TREATMENT, 182, 194-207., 3. Jha, R., Rao, M. D., Meshram, A., Verma, H. R., & Singh, K. K. (2020). Potential of polymer inclusion membrane process for selective recovery of metal values from waste printed circuit boards: A review. Journal of Cleaner Production, 121621.)
R: I’m not agreeing with this comment, because this manuscript concerns simple kinetic models which can be used for maximum flux calculation. The history, benefits (and drawbacks), and ion transport mechanisms will be (I hope) detailed described in other manuscript which was accepted to publish in Special Issues of Membranes i.e. Polymer Inclusion Membranes. Moreover, proposed by the Reviewer #2 reference No1 concern an electrodialysis process in which PIMs were used, not a carrier-mediated transport. I also would like to explain, that in the case of other proposed references, I have no access to PDF file whereas from abstract section only general information is provided.
- More explanations about the effect of PIM composition on the final results are required.
R: Since membranes of the same composition were used in experiments, I cannot respond to this comment in the manuscript. However, additional experiments which were carried out for D2EHPA and reactive ionic liquids as carriers confirm conclusions that model based on equation similar to the first-order chemical reaction equation (proposed by Danesi) can be applied only for description of the transport kinetics for substances with the highest selectivity (highest flux) and/or for permeability coefficients calculated from selected initial points. Whereas the model based on equation similar to first–order chemical reaction equation with equilibrium, is more flexible and can be applied for all of the transported substances (ions) regardless of selectivity.
- The PIM components (polymer, carrier, and the plasticizer) how have been selected for these experiments? If it has been selected based on a specific reference, please mention it.
R: Some additional explanations were added i section 2.1. (marked in blue):
“PIMs were prepared as flat membranes by a solution casting and solvent evaporation technique at room temperature [18]. A solution of CTA, carrier and the plasticizer in dichloromethane (Riedel–de Haën, purity ³99.9%) were prepared and poured into a Petri dish (7 cm diameter). After evaporation of the organic solvent (24 h) the membrane was immersed for 12 h in distilled water to swell and used in a transport of metal ions. The composition of the membrane was the same for all of the experiments i.e.: 43 wt. % D2EHPA, 19 wt. % NPOE and 38 wt. % CTA. Such composition is typical in studies carried out by other authors [19, 20]. The total membrane weight after solvent evaporation equaled (0,312±0.008) g and its thickness was (0.0568±0.0042) mm.”
- What is “partition coefficient” mentioned in page 2? More explanation is required.
R: The partition coefficient is the term taken from P.R. Danesi work. However, because IUPAC Gold Book not recommended this term, it was change into distribution constant (synonymous with partition ratio).
- In section “2.Experimental”, the range of initial ion concentration in feed phase is in the range of 0.0005 to 0.01 M; but the initial ion concentrations showed in Fig. 2 is 2 M!!!! Please explain this discrepancy.
R: In Fig. 2, the units of the dependent variable (vertical axis) are in [mmol/dm3] it means that initial concentration equals 0.002 M.
- In section 2.1, total weight of the casted membrane, as well as each PIM components weight must be mentioned.
R: Section 2.1 was changed (changes are marked in blue):
Reviewer 3 Report
Review report on the manuscript titled:
Some critical remarks about mathematical model used for description of transport kinetics in polymer inclusion membrane systems
- The carrier agent used is classical and the transport of the divalent ions does not clearly show the positive effect of the model based on equilibrium. While the transport of organic molecules is better suited to this model
- Explain how to maintain an acidity of 0.5 M in the receiving phase and neutral distilled water in the feed phase? Diffusion of H + ions is possible !!!!
- The type of PIM prepared and used is the least permeable among the different types of known PIMs (Bibliography).
- It is necessary to divide Ji by the thickness of the membrane, in order to be able to compare the fluxes for membranes of different thicknesses.
- The device in Figure 1 does not allow complete extraction of Pb(II) and Zn(II) ions (Figures 2 and 5), especially since the author talks about the model based on equilibrium between the two compartments. Justify and explain these results.
- The obtained permeabilities (Table 1) are too high for the announced values of the initial fluxes. Justify these results and values.
- The model based on equilibrium of the substrate between the two compartments of the cell is the closest to reality because the substrate diffuses in both directions until an equilibrium state (equal concentrations in the two phases). This model has been adopted and used since 2000 by Verchère et al and Hlaibi et al. And from the relation (9) of the manuscript, it is easy to find exactly the relation described and used by these authors by replacing Csinf by C0 / 2 and Cf (t = 0) by C0 provided that Vf = Vs.
- In addition, these authors were able from this model based on reversible diffusion to determine two important parameters Kass and D* relating to the movement of the substrate in the membrane phase.
- Therefore, the author should review and improve the experimental and bibliographic sections.
Author Response
RESPONSE TO COMMENTS
Reviewer #3
- The carrier agent used is classical and the transport of the divalent ions does not clearly show the positive effect of the model based on equilibrium. While the transport of organic molecules is better suited to this model
Response: The applied in the experiments D2EHPA is a typical carrier for coupled counter-transport of divalent ions, which was confirmed by the results of other authors. In such a systems a coupled-counter transport mechanism is observed. In a relative simple systems (only one metal cation in the feed solution), the concentration vs. time dependences can be successfully described by the model proposed by P.R. Danesi. In the case of multicomponent solution (or in systems in which transport efficiency is relative low) the concentration vs. time dependences indicate that the system tends to some kind of equilibrium (even after a very long transport time for Vf = Vs, the concentration of ions in the feed solution does not decrease to zero as well as the concentration of the stripping solution does not equal the initial concentration of the feed solution). Such type of relationship can be quantitatively described be the model proposed in this manuscript.
In the case of an organic substance transport (non-ionic), a carrier-mediated transport (facilitated diffusion but not coupled-counter transport) occurs and therefore much simpler equation can be used (see response for comment 7). In the case of an organic substance which can reacts immediately in the stripping solution, creating a product which is not transported through the membrane, the transport can be described by Eq.(3).
- Explain how to maintain an acidity of 0.5 M in the receiving phase and neutral distilled water in the feed phase? Diffusion of H + ions is possible !!!!
R: Diffusion of H+ is a typical phenomenon observed for a coupled-counter transport. There is no simple solution to the problem associated with the change in pH of the feed solution. The use of a pH stat titration system leads to the introduction of additional ions into the feed solution. This ions can also be transported by the carrier and reducing the flux of other ions. In the case of the stripping solution, the change in the concentration is low (from 0.5 M to 0.484M), and only when all the metal ions were transported to the stripping solution. It should be pointed out that most of membrane transport experiments of other researchers are carried out under similar conditions.
- The type of PIM prepared and used is the least permeable among the different types of known PIMs (Bibliography).
R: I’m not agreeing with this comment. The results indicated that maximum fluxes (from 2x10-11 up to 9x10-10 mol/cm2s) are similar to the values observed by other authors in systems operating under similar conditions [19,20].
- It is necessary to divide Ji by the thickness of the membrane, in order to be able to compare the fluxes for membranes of different thicknesses.
R: The mean thickness of the membranes was given in section 2.1 according to the Reviewer #1 expectations.
- The device in Figure 1 does not allow complete extraction of Pb(II) and Zn(II) ions (Figures 2 and 5), especially since the author talks about the model based on equilibrium between the two compartments. Justify and explain these results.
R: The results presented in Fig.2 and 5 indicated almost complete extraction of Zn(II) and Pb(II) (recovery factor>95%). Probably if the transport time would be longer, the maximum recovery factor will be achieved. However, due to change in pH in the feed solution (H+ counter-transport) and metal concentration change in the feed and stripping solution (which reducing the driving force of the process) the distribution constant at the respective interphases are time dependent and finally decreases the transport efficiency. Moreover some discrepancy can results from uncertainty of the metal ion concentration analysis (the maximum relative error c.a.3%).
- The obtained permeabilities (Table 1) are too high for the announced values of the initial fluxes. Justify these results and values.
R: Thank you for checking the values carefully. Indeed in the first row, the maximum flux equals 3.4x10-10. However in the case of the rest of the results, some inaccuracy results from rounding of numerical values according to the rules of significant digits recording.
- The model based on equilibrium of the substrate between the two compartments of the cell is the closest to reality because the substrate diffuses in both directions until an equilibrium state (equal concentrations in the two phases). This model has been adopted and used since 2000 by Verchère et al and Hlaibi et al. And from the relation (9) of the manuscript, it is easy to find exactly the relation described and used by these authors by replacing Csinfby C0 / 2 and Cf (t = 0) by C0 provided that Vf = Vs.
R: The model described in mentioned papers, concern carrier mediated diffusion through supported liquid membrane. Only in such type of transport Csinf=C0/2 for Vf=Vs i.e. the maximum concentration in the stripping solution is a half of the initial feed solution concentration (recovery factor=50%). In the case of carrier-mediated counter transport (e.g. with D2EHPA as a carrier) an active transport is observed in the case when Vf>Vs. When Vf=Vs, the maximum concentration of the transported substance in the striping solution may be equal to the initial feed solution concentration (recovery factor=100%).
- In addition, these authors were able from this model based on reversible diffusion to determine two important parameters Kassand D* relating to the movement of the substrate in the membrane phase.
R: Due to the above-described differences between these models, the calculation of this two important parameters i.e. Kass and D* is not possible.
- Therefore, the author should review and improve the experimental and bibliographic sections.
R: I hope that all of my responses and changes in the manuscript have satisfied the Reviewer #3 expectations.
Round 2
Reviewer 2 Report
The manuscript titled “Some critical remarks about mathematical model used for description of transport kinetics in polymer inclusion membrane systems” by Piotr SzczepaÅ„ski, while Authors employed some revision on the manuscript, but still, they did not reply to some of the main concerns about this work. In this way, for final decision on this work more clarification is required.
- The abstract should be revised by reporting more numerical values.
- More explanations about the effect of PIM composition on the final results are necessary.
- The PIM components (polymer, carrier, and the plasticizer) how have been selected for these experiments? If it has been selected based on a specific reference, please mention it.
Author Response
RESPONSE TO COMMENTS
Reviewer #2
- The abstract should be revised by reporting more numerical values.
Response: According to the Reviewer #3 expectations the following sentences were added (changes marked in green):
Page 1 lines 10-16
The models were fitted to the real experimental data of Zn(II), Cd(II), Cu(II), and Pb(II) simultaneous transport through PIM with di-(2-ethylhexyl)phosphoric acid (D2EHPA) as a carrier, o–nitrophenyl octyl ether (NPOE) as a plasticizer, and cellulose triacetate (CTA) as a polymer matrix. The selected membrane was composed of 43 wt. % D2EHPA, 19 wt. % NPOE and 38 wt. % CTA. The results indicated that the calculated initial fluxes (from 2´10-11 up to 9´10-10 mol/cm2s) are similar to the values observed by other authors in systems operating under similar conditions.
Which I hope fulfill the Reviewer #2 expectations.
- More explanations about the effect of PIM composition on the final results are necessary.
R: According to the Reviewer #3 expectations the following explanation was added:
Page 9 lines 246-255:
The value of the maximum flux depends on many operational conditions e.g. the concentration and the type of the feed and the stripping solution. In a particular way this value depends on the membrane composition, its thickness, and type of the carrier. In order to determine the influence of the membrane composition on the concentration vs time dependence, several additional experiments were carried out. The results confirm that the model proposed by Danesi [1,2] can be applied only for description of the transport kinetics for substances with the highest selectivity (i.e. the highest flux). Whereas the model represented by Eqs (8) and (9) is more flexible and can be applied for all of the transported substances (ions) regardless of the selectivity. This topic, however, is beyond the scope of this manuscript and will be considered and discussed in the subsequent manuscript.
- The PIM components (polymer, carrier, and the plasticizer) how have been selected for these experiments? If it has been selected based on a specific reference, please mention it.
R: It was mentioned in the manuscript that “Such composition is typical in studies carried out by other authors [19, 20].” However, in order to fulfil the Reviewer #3 expectation, the above sentence was changed into:
Page 4 lines 122-123:
The concentrations of D2EHPA and NPOE selected for the experiments correspond to the maximum fluxes of ions observed by other authors in the similar systems [19, 20].
Reviewer 3 Report
The author did not respond to my criticisms especially the following three points:
- Improve the bibliography section by comparing this work with that of authors who have already worked on a model based on an equilibrium between the two phases.
- How to maintain a difference in acidity between the two compartments especially for a long time for the processes carried out ?
- For a closed circuit system with two compartments of equal volumes, how the author was able to achieve extraction rates greater than 50% ?
Author Response
RESPONSE TO COMMENTS
Reviewer #3
The author did not respond to my criticisms especially the following three points:
1) Improve the bibliography section by comparing this work with that of authors who have already worked on a model based on an equilibrium between the two phases.
Response: In the literature, I found only one example of the application of the model represented by Eqs (8) and (9) i.e. Yoshida, S.; Hayano, S. Kinetics of partition between aqueous solutions of salts and bulk liquid membranes containing neutral carriers. J. Membr. Sci. 1982, 11, 157–168; https://doi.org/10.1016/S0376-7388(00)81398-8, which was cited in the manuscript as [17].
Probably Reviewer #3 still does not distinguish between the model (simple carrier-mediated transport, i.e. a simple diffusion mediated by a carrier) which is based on equilibrium between the phases and the proposed model (for carrier-mediated coupled transport) in which the concentration in the feed and the stripping solution is allowed to approach equilibrium. I realized that the description of the proposed model might be incomprehensible. Therefore some changes were added (marked in green):
Page 2, lines 74-77:
Taking into account the volume of the feed (Vf) and stripping (Vs) solution, the membrane surface area (A), and assuming that the concentration in the feed and the stripping solution is allowed to approach equilibrium, the respective flux equation derived from Eq.(5) can be integrated to:
Page 7, lines 199-201:
Nevertheless, the fit of the model, especially for cf = f(t) plot and time higher than 40 h, is still unsatisfactory and suggests that the concentration is reaching the equilibrium.
Page 9, lines 248-249:
... especially in the case when cf inf or cs inf values are experimentally inaccessible i.e. if the concentration in the respective phases has not reached the equilibrium.
I would like to point out that, the observed equilibrium (in concentration) can be interpreted similar to the Gibbs-Donnan equilibrium in which the distribution of permeable charged ions are influenced by both their valence and the distribution of non-diffusable ions, such that at equilibrium the products of the concentrations of paired ions on each side of the membrane will be equal. The ion transport (omitting the carrier contribution, which acts similar to an ion exchange membrane) can be described by:
M2+(f)+2H+(s) ⇄ M2+(s)+2H+(f)
The driving force is defined by:
Eq (1r)
If the equilibrium is attained ( the following relation is obtained:
Eq (2r)
Therefore:
Eq (3r)
K denote the Donnan equilibrium constant. In a typical ion-exchange transport K<<1. In the case of simple diffusion (or carried mediated simple diffusion), Cf=Cs and this value is equal 1 (for Vf=Vs).
2) How to maintain a difference in acidity between the two compartments especially for a long time for the processes carried out ?
R: Difference in acidity is mainly maintained by the application of acid concentration much higher than the initial metal concentration in the feed solution (Cs0(acid)=0.5M, Cf0(metals)=0.002M).
3) For a closed circuit system with two compartments of equal volumes, how the author was able to achieve extraction rates greater than 50% ?
R: Because the transport occurs according to the carrier-mediated coupled counter-transport mechanism in which ions cross the membrane in opposite directions (and fluxes are coupled). Energy for metal ion uphill transport is gained from the coupled transport of protons in the opposite to metal ions from the stripping to the feed solutions (similar equation for DG as in the Donnan dialysis). It means that concentration of protons (much higher than initial metal ions concentration) is the driving force of the transport. The recovery factor (RF=(cf0-cf)/cf0*100%) can reach values close to 100%. This can be easily calculated even from Eq. (3) that for A=17 cm2, V=200 cm3, P=2x10-4cm/s, cf0=0.002M, and t=200000 s (i.e. 55.55h) the concentration in the feed solution equal 6.67x10-5 M, which leads to RF=96.7%.

Round 3
Reviewer 2 Report
Now, the paper can be published in the present format.
Author Response
RESPONSE TO COMMENTS
Reviewer #2
- English language and style are fine/minor spell check required
Response: Minor spell check was done – see List of changes.
Reviewer 3 Report
Equations 8 and 9 are the same equations already described by other authors by changing the notations Cf (inf) patr C0 / 2 because at an infinitely large time, there will be charge equilibrium between the H+ and the substrate ions in the two compartments.
For question 3, the answer is not convincing, several studies have shown that the transport is conducted by the interaction between the metal ions and the carrier and not by the concentration of H + ions in the receiving compartment
Author Response
RESPONSE TO COMMENTS
Reviewer #3
- Equations 8 and 9 are the same equations already described by other authors by changing the notations Cf (inf) patr C0 / 2 because at an infinitely large time, there will be charge equilibrium between the H+ and the substrate ions in the two compartments.
Response: According to the Reviewer #3 expectations some explanation was added. The appropriate references [19-21] was inserted. Examples of the application of the Eq. (11) quoted by the Reviewer were limited only to the PIM system:
“Eq. (8) can be easy simplified to the relationship presented by Eq (3) by assuming that the concentration of a substance in the feed solution at infinite time goes to zero (cfinf → 0). Another simplification of Eq (8) can be obtained assuming that Vf = Vs and cfinf = cf,t=0/2. Such conditions indicate the membrane system in which a simple diffusion or carrier–mediated simple diffusion process occurs. The respective equation takes the following form:
Eq. (11)
in which Pd denote the diffusive permeability coefficient [cm/s]. For instance, Eq (11) was applied for characterization of NaCl diffusive permeability through PIM containing Aliquat 336 [19] or organic acids transport across PIM in which 1-alkylimidazols and TOA were applied as a carrier [20,21].”
- For question 3, the answer is not convincing, several studies have shown that the transport is conducted by the interaction between the metal ions and the carrier and not by the concentration of H + ions in the receiving compartment
R: I hope that the additional explanations (about simple diffusion and carrier-mediated simple diffusion processes) finally fulfill the Reviewer #3 expectations.